# Therapeutic Effect of Intense Pulsed Light in Patients with Sjögren’s Syndrome Related Dry Eye

**DOI:** 10.3390/jcm11051377

**Published:** 2022-03-02

**Authors:** Yanan Huo, Qi Wan, Xinzhu Hou, Zhiyong Zhang, Jinchuan Zhao, Zhiyi Wu, Xiuming Jin

**Affiliations:** 1Eye Center, The Second Affiliated Hospital of Zhejiang University School of Medicine, Hangzhou 310009, China; 201911122711389@zcmu.edu.cn (Q.W.); zhangzhiyong@zju.edu.cn (Z.Z.); wuzhiyi0501@zju.edu.cn (Z.W.); 2Second Clinical Medical School, Zhejiang Chinese Medical University, Hangzhou 310053, China; 202011122711619@zcmu.edu.cn; 3Zhejiang Institute of Medical Device Supervision and Testing, Hangzhou 310018, China; zhaojinchuan@mdst.org.cn

**Keywords:** Sjögren’s syndrome, intense pulsed light, dry eye, keratoconjunctivitis sicca, filamentary keratitis

## Abstract

This prospective randomized study evaluated the efficacy and safety of intense pulsed light (IPL) and meibomian gland expression (MGX) as polytherapy for Sjögren’s Syndrome-related dry eye (SS-DE). The study enrolled 55 participants with SS-DE, 27 for the treatment group and 28 for the control group. The treatment group underwent three IPL-MGX treatments, three weeks apart. A randomly-selected eye from each patient was assessed at baseline and on weeks 9, 12, and 15 for Snellen best-corrected visual acuity (BCVA), intraocular pressure, Ocular Surface Disease Index (OSDI) score, conjunctival congestion, tear meniscus height, non-invasive tear breakup time (NBUT), Schirmer’s I test (SIT), corneal fluorescein staining (CFS), meibomian gland (MG) dropout, eyelid margin abnormality, MGX and meibum quality. OSDI, NBUT, CFS, MGX, and meibum quality were significantly improved in both groups, particularly in the treatment group. The eyelid margin abnormality improved significantly in the treatment but not in the control group on weeks 12 and 15. Snellen BCVA, conjunctival congestion, and SIT improved significantly in the treatment group, but the two groups were statistically similar. Our results indicated that three IPL-MGX sessions could significantly improve the subjective and objective characteristics of SS-DE, representing a promising treatment strategy.

## 1. Introduction

Sjögren’s syndrome (SS) is an autoimmune disease characterized by the lymphocytic infiltration of the moisture-producing glands, including the sebaceous, sweat, salivary, and lacrimal glands, resulting in its two most common symptoms: dry eyes (DE) and a dry mouth [1,2]. The disease might have other concomitant systemic displays [3]. The SS-associated DE (SS-ED) severely affects the patients’ quality of life and activity range. While SS-DE might have a range of ocular presentations, the most common is keratoconjunctivitis sicca (KCS). Patients with SS-DE show significantly more serious signs and symptoms, including poorer and more blurred vision than those with non-SS DE [4].

Traditionally, DE was conveniently divided into the aqueous-deficient and evaporative subgroups. These two DE subgroups differ in the pathophysiological and background aspect of their DE. Patients with SS are classified exclusively into the aqueous-deficient DE subgroup because their damaged lacrimal glands secrete less of the tears’ aqueous portion, but they might also present meibomian gland (MG) dysfunction (MGD) [5]. Therefore, patients with SS should be treated for both DE subtypes.

The treatment for SS-DE depends on the severity of the symptoms. Artificial tears or lubricants are sufficient at the early stage of the disease [6]. Patients with a progressive or more severe disease might need topical corticosteroids [7] or cyclosporine [8]. Other treatment options include dietary supplements of omega-3 essential fatty acids [9] and the use of punctal plugs [10]. SS, particularly progressive SS, remains difficult to manage despite the various available treatment options. Some patients with SS might present with DE resistant to the treatment options mentioned above. The most severe cases, particularly those unresponsive to topical corticosteroids or cyclosporine, could be considered for applying topical autologous serum or partial tarsorrhaphy to reduce environmental exposure and evaporation [11].

Intense pulsed light (IPL) has been widely used to treat evaporative DE, mostly secondary to MGD [12,13] or ocular demodicosis [14]. As far as we know, only one retrospective study has reported using IPL to treat seven patients with SS-DE [15]. However, that previous study was not randomized or performed in a masked manner. Therefore, this is the first study to evaluate the therapeutic effects and safety of IPL and MG expression (MGX) as polytherapy for SS-DE.

## 2. Materials and Methods

### 2.1. Subjects

This study followed the tenets of the Declaration of Helsinki. The Ethics Committee of The Second Affiliated Hospital of Zhejiang University School of Medicine approved the study (No. 2019-270). The study was registered as a clinical trial on 9 August 2020 (ChiCTR2000035344). All patients provided written informed consent before enrollment in this research.

Rheumatologists diagnosed SS in all participants following the Sjogren’s International Collaborative Clinical Alliance (SICCA) classification criteria from 2012 [16]. These criteria comprise (1) serum positive for anti-SSA/Ro or anti-SSB/La, or positive rheumatoid factor and antinuclear antibody titer ≥ 1:320; (2) histological assessment showing labial salivary gland infection with lymphocytic focus score ≥ 1 per 4 mm^2^; (3) a score of ≥3 for KCS during corneal staining. Refractory SS-DE was determined when a patient failed to respond to at least three standard treatment approaches through management for one year or more. These treatments included lubricant eye drops, topical ointment, warm compress and massage of the eyelid, topical anti-inflammatory therapy, moisture chamber spectacles, and for those not currently using-contact lenses, punctal plugs and systemic immunosuppression.

### 2.2. Experimental Design

We randomly assigned patients with SS-DE aged 18-70 years to the IPL-MGX treatment or control group. Patients in the IPL-MGX group underwent three IPL-MGX treatments three weeks apart and three follow-up assessments 9, 12, and 15 weeks from the start of treatment. All participants were prescribed sodium hyaluronate eye drops four times daily during the study and follow-up. The patients were allowed to continue with home care such as warm compresses and lid hygiene.

A day before the first treatment and at each follow-up assessment, the treatment efficiency was evaluated using Snellen best correct visual acuity (BCVA), intraocular pressure (IOP), Ocular Surface Disease Index (OSDI) score, conjunctival congestion, non-invasive breakup time of tear film (NBUT), Schirmer’s I test (SIT), tear meniscus height (TMH), corneal fluorescence staining (CFS), MG dropout, eyelid margin abnormalities, MGX, and meibum quality.

CFS was considered the primary outcome measure, while the secondary outcome measures included OSDI score, TBUT, MG dropout, eyelid margin abnormalities, MGX, meibum quality, SIT, TMH, and conjunctival congestion. The treatment safety was assessed by Snellen BCVA, IOP and slit lamp examination. In this investigation, one eye from each patient was randomly chosen and evaluated.

### 2.3. IPL-MGX Treatment

The M22 IPL system (Lumenis, Tel Aviv, Israel), set for the AOPT mode, was used in this research. The energy parameters were estimated based on Fitzpatrick skin type and the patient’s tolerance and comfort (density, 15–17 J/cm^2^). Each patient received two perpendicular IPL treatments, once from each side. The treatment from the right included the cheeks and nose, while the one from the left extended the interior margin of the protective eye shields. An Aritia Meibomian Gland Compressor (Katena Product Inc., Denville, NJ, USA) was used to perform the MGX on the upper and lower eyelids immediately after the IPL therapy. The patients in this group were given 0.4% oxybuprocaine hydrochloride eye drops (Santen Pharmaceutical Co., Ltd., Osaka, Japan) for pain relief through the procedure.

An additional inclusion criterion was Fitzpatrick skin types of 1–4 [17]. Patients were excluded if they had implants in the treatment area; cosmetic eyelid surgery in the last five years; glaucoma; graft-versus-host disease, autoimmune connective tissue diseases other than rheumatoid arthritis or lupus; acute solar dermatitis; allergic disease; had eye surgery less than three months earlier; any physical or mental condition interfering with successful participation in the study; recent LipiFlow or MGX treatment.

### 2.4. Clinical Assessment

#### 2.4.1. Ocular Surface Disease Index (OSDI)

The clinical parameters evaluated were similar to our previous study [14]. The 12-item OSDI questionnaire assessed the ocular surface-related symptoms and their severity and frequency. The sum of all 12 OSDI items would result in a score of 0–100 points.

#### 2.4.2. Schirmer’s I Test (SIT)

The SIT was performed without anesthesia. A sterile dry strip of filter paper (Jingming New Technological Development Co., Ltd., Tianjin, China) was placed inside the inferior fornix for 5 min. The length of the moistened area on the strip determined the lacrimal gland functionality.

#### 2.4.3. Tear Meniscus Height (TMH) and Conjunctival Hyperemia

The lower TMH and conjunctival hyperemia were assessed automatically by the OCULUS Keratograph 5M (OCULUS Optikgeräte GmbH, Wetzlar, Germany).

#### 2.4.4. Tear Film Stability

NBUT was measured using the OCULUS Keratograph 5M (OCULUS Optikgeräte GmbH, Wetzlar, Germany). We asked the patients to blink several times before leaving their eyes open without blinking. We measured the time in seconds from the last complete blink to the first disturbance or irregularity of the concentric rings reflected on the cornea surface. The average NBUT of three tests was recorded.

#### 2.4.5. Corneal Fluorescein Staining (CFS)

CFS was performed with a moist fluorescein strip (Jinming New Technological Development Co., Ltd., Tianjin, China). The cornea was divided into five sections (superior, temporal, nasal, inferior, and central), and a staining severity was assigned to each, ranging from 0 (no staining) to 3 (severe).

#### 2.4.6. Meibomian Gland (MG) Morphology

Meibography was rated by the OCULUS Keratograph 5M (OCULUS Optikgeräte GmbH, Wetzlar, Germany) on a scale of 0–3: 0, no gland loss; 1, <1/3 lost area; 2, area of loss between 1/3 and 2/3; and 3, >2/3 lost area.

#### 2.4.7. Eyelid Margin Abnormalities

The eyelid margins were assessed for abnormalities on a 0–4 scale following these four criteria: irregular eyelid margins, vascular engorgement, clogged MG ducts, and displaced mucocutaneous junction.

#### 2.4.8. Meibomian Gland Expression (MGX)

The MG Evaluator assessed the MGX on the on lower tarsal plate. The eyelid was divided into three sections (nasal, central, and temporal). Each section contained five glands. We counted the glands expressing meibum in each section and rated the expression as follows: 0, normal; 1, 3–4 glands expressed; 2, 1–2 glands expressed; 3, no gland expression. The total MGX score could be in the range of 0–9.

#### 2.4.9. Meibum Quality

Each of eight MGs in the center of the lower eyelid was rated on a scale of 0–3: 0, no secretion; 1, inspissated, toothpaste-like fluid; 2, viscous, opaque, or yellow fluid; 3, clear fluid (total score range, 0–24).

#### 2.4.10. Safety Assessments

Treatment safety was evaluated by the Snellen BCVA, IOP, and slit-lamp examination performed at the start of the study and during each follow-up session.

### 2.5. Statistical Analysis

A statistical analysis was performed using IBM SPSS Statistics for Windows, Version 25.0 (IBM Corp., Armonk, NY, USA). We performed a sample size calculation following the method used in previous studies [18,19,20]. We hypothesized that the CFS results would differ by 25% between the IPL-MGX and control groups. Based on this assumption, a power of 90%, and a two-sided significance level of 0.05, our study needed 24 patients in each group. Only data of patients completing the entire study were analyzed. Continuous variables are presented as means ± standard deviations (SDs). The Shapiro-Wilk test assessed these variables for normal distribution. The paired-samples *t*-test or paired Mann-Whitney *U* test compared continuous variables between baseline and the follow-up assessments. The control and IPL-MGX groups were compared by the independent-sample *t*-test or Mann-Whitney *U* test at baseline and after the treatment period. Statistical significance was set at *p* < 0.05.

## 3. Results

This prospective randomized controlled trial enrolled 55 patients with SS upon diagnosis with bilateral DE. One patient in the IPL-MGX group could not adhere to the follow-up schedule due to the COVID-19 quarantine policy and left the study. Four of the control group patients left the study because they saw no relief in their DE symptoms. All analyses was based on the remaining participants. Figure 1 presents an outline of the study and its stages (recruitment, withdrawal, treatment sessions, and follow-up assessments).

Fifty patients, one male (2%) and forty-nine females (98%), completed the study. The IPL-MGX group was comprised of 26 patients, a male and 25 females, aged 53.46 ± 10.71 (range, 32–78) years. The control group was comprised of 24 patients, all female, aged 51.71 ± 13.32 (range, 30–67) years. The groups were similar in age, DE, and SS medical history. Refractory SS-DE occurred in 23 patients (88.46%) in the treatment group and 21 (87.5%) in the control group. As shown in Table 1, the groups were similar in their patient characteristics.

The groups were compared for the parameters assessed during each visit (Table 2). The time courses of Snellen BCVA, ODSI, NBUT, CFS, eyelid margin abnormality, MGX and meibum quality are presented in Figure 2, Figure 3 and Figure 4. The groups were similar in all parameters at baseline. The Snellen BCVA score at three follow-up assessments was lower than at baseline in the IPL-MGX group (all *p* ≤ 0.001), while the control group remained unchanged (all *p* > 0.05; Figure 2A). The OSDI scores were lower than at baseline in all three follow-up assessments in both groups (all *p* < 0.05). All three follow-up scores in the IPL-MGX group were considerably lower than in the control group (all *p* < 0.001; Figure 2B). Compared to baseline, the two groups showed higher NBUT (Figure 3A) and lower CFS (Figure 3B) at all follow-up assessments (all *p* < 0.05). The improvement in NBUT and CFS in the treatment group was greater than in the control group when assessed on weeks 12 and 15 (all *p* < 0.05). Compared to baseline, a significant decrease in eyelid margin abnormalities (Figure 4A) and a substantial increase in MGX (Figure 4B) and meibum quality (Figure 4C) were apparent at all three follow-up sessions in the IPL-MGX group (all *p* > 0.05). Significant improvements in the MGX and meibum quality were also noted at weeks 12 and 15 in the control group (all *p* > 0.05). The eyelid margin abnormalities remained unchanged throughout the study in the control group (all *p* > 0.05). The eyelid margin abnormalities, MGX and meibum quality in the IPL-MGX group differed significantly from those in the control groups (all *p* > 0.05), except for MGX at the week 15 assessment (*p* = 0.094).

We found similar Snellen BCVA, conjunctival congestion, TMH, and SIT at weeks 9, 12, and 15 in the two groups. MG dropouts did not change over the four visits in either group (all *p* > 0.05).

The KCS-specific symptoms have improved following the IPL-MGX treatment. Figure 5 and Figure 6 show two representative cases of patients with SS-DE in the IPL-MGX group who exhibited significant improvement in their ocular surface conditions between baseline and the final follow-up visit. Both patients had notable ocular surface inflammation, poor vision, and could barely open their eyes before treatment (Figure 5A–D and Figure 6A–D). Case 3 (Figure 5) was a 60-year-old female with SS for 14 years and DE for 13 years. The patient was previously treated with topical and oral corticosteroids, topical cyclosporine, and autologous serum, but none achieved disease progression control. The patient showed severe symptoms including strong foreign body sensation, burning, stinging, photophobia, blurred vision, and ocular pain. Case 11 (Figure 6) was a 36-year-old female with SS for 7 years and DE for 5 years. The patient had similar ocular irritation symptoms as in case 3. Previously treatments with topical corticosteroids, cyclosporine, artificial tear eye drops, warm compresses combined with MGX, and eyelid hygiene had failed. Reduced filamentary keratitis, corneal epithelial healing, and improved vision were noted at last follow-up in both cases (case 3: Figure 5E–H, case 11: Figure 6E–H).

## 4. Discussion

SS is a chronic, currently incurable, and potentially deadly disease. Cell-mediated immunity plays an important role in the pathogenesis of SS-DE [3]. Although it is traditionally classified as aqueous tear-deficient DE, mounting evidence suggests that ocular surface inflammation is not limited to the lacrimal glands [21]. An inflammation cascade is initiated on the ocular surface by lymphocytic infiltration and the increase in cellular inflammatory mediators. This cascade might cause an imbalance among the components of the complex ocular system that include the eyelid, tears, mucous, and epithelial surfaces. We believed the inflammation of the ocular surface and exocrine glands, and the diminished neural innervation, could further impair the accessory lacrimal glands, corneal epithelial cells, goblet cells, and MGs. These damages result in deficiency of all the tear film constituents, leading to the observed presentation that is often more severe than in non-SS DE. Unlike those with non-SS DE, patients with SS-DE were reported to have poorer vision and quality of life [5].

Controlling inflammation may play an important role in treating SS-DE. However, the management of SS-DE in clinical practice could be challenging for ophthalmologists. Topical corticosteroids, cyclosporine, and even autologous serum often achieve only temporary and partial improvement, possibly because this complex pathology is only partially understood, resulting in symptom-oriented treatment rather than the targeting of the cause. Most patients in our study (IPL-MGX group: *n* = 23, 88.5%; control group: *n* = 21, 87.5%) had refractory SS-DE before recruitment. Patients with moderate to severe ocular surface conditions, such as persistent filamentary keratitis and corneal epithelium defect, might not respond well to conventional anti-inflammatory medication and may require additional measures.

As far as we are aware, only one previous IPL study was performed, and that study included only a few patents with SS-DE (seven in the IPL-MGX treatment group and six in the control group) [15]. That study had several limitations. First, both eyes of each participant were included, which, statistically, is not considered best practice. Second, these thirteen patients were recruited from three centers with no detailed and consistent diagnostic criteria such as serological testing or labial salivary gland biopsy. Third, all patients were allowed to continue their various ocular medications during the study. Moreover, there might be a high risk of bias, as it was not randomized or performed in a masked manner. All these important confounding factors reduce the confidence in the reported results.

Our study was the first prospective randomized controlled trial to demonstrate the beneficial effects of the IPL-MGX treatment in patients with SS-DE. The patients in our study were randomly assigned to the IPL-MGX or the control group. The treatment group patients received three IPL-MGX treatments at three-week intervals. All characteristics were assessed at baseline and on weeks 9, 12, and 15 after the first treatment. Some of the measured parameters have improved through follow-up in both study groups, but the IPL-MGX groups showed greater improvement than the control group in the OSDI score, NBUT (weeks 12 and 15), CFS (weeks 12 and 15), eyelid margin abnormalities, MGX (weeks 9 and 12), and meibum quality. These findings were consistent with our earlier studies showing the effectiveness of IPL as a treatment for MGD and ocular demodicosis [14,22].

The Snellen BCVA score was included for safety assessment when we designed this experiment. Unexpectedly, the vision of participants treated with IPL-MGX showed a remarkable improvement that persisted through all three follow-up assessments (all *p* < 0.001), unlike the patients with non-SS DE in our previous IPL experiment [14]. Although the differences between the treatment and control groups did not reach the significance level, we were satisfied with the significant improvement in visual acuity in some patients, including those in which topical cyclosporine or autologous serum treatment had failed. The lower CFS and improved vision might suggest that the IPL treatment enhanced the integrity of the ocular surface. The decreased inflammation and improved ocular surface could initiate a cascade that includes improved corneal sensory nerve ending activation, enhanced neural signals to the lacrimal gland, and improved tear secretion. These could result in the improvement seen in SIT and TMH in this study.

Although the mechanism of the IPL treatment remains unclear, we believe that the main reason for the improved vision in the patients with SS is that IPL reduced the ocular surface inflammation considerably, lowering apoptosis of the corneal epithelial and conjunctival goblet cells and filamentary keratitis, and improving the MGD symptoms and signs. Similar patterns were observed for conjunctival congestion and SIT, with IPL-MGX leading to a significant improvement over baseline not seen in the control group. However, the differences in these parameters between the two groups were statistically insignificant in this study (all *p* > 0.05). Our findings suggest that IPL could improve vision and reduce ocular surface inflammation in patients with SS-DE.

This study had several limitations. We used an IPL protocol for MGD-related DE (three IPL-MGX treatment sessions at three-week intervals). However, it seems that unlike patients with non-SS DE, those with SS and severe ocular surface damage might need more than three IPL treatment sessions or shorter inter-treatment intervals. Moreover, the most severe cases may require combination therapy that includes topical corticosteroids or cyclosporine eyedrops. Therefore, a further long-term study with a combination therapy study is necessary to better understand the effectiveness and mechanism of the IPL-MGX treatment.

## 5. Conclusions

In conclusion, our study showed IPL-MGX to be a promising treatment regimen for patients with SS-DE, especially those refractory to the standard therapies. Further studies might promote other therapeutic possibilities; however, our findings indicate that IPL-MGX significantly improves the objective and subjective disease characteristics in patients with SS-DE.

## Figures and Tables

**Figure 1 jcm-11-01377-f001:**
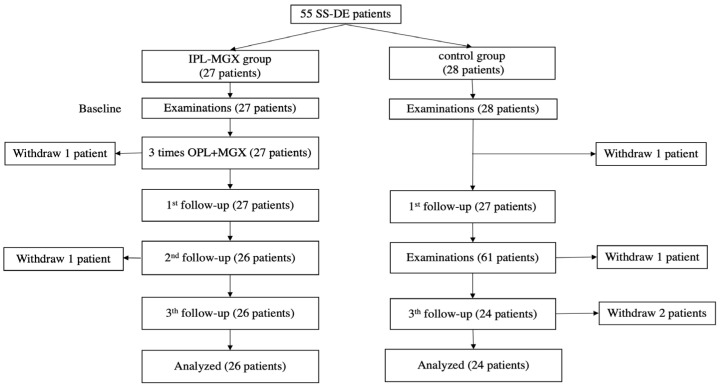
The treatment protocol and follow-up schedule for the intense pulsed light (IPL)-meibomian gland expression (MGX) and control groups are presented.

**Figure 2 jcm-11-01377-f002:**
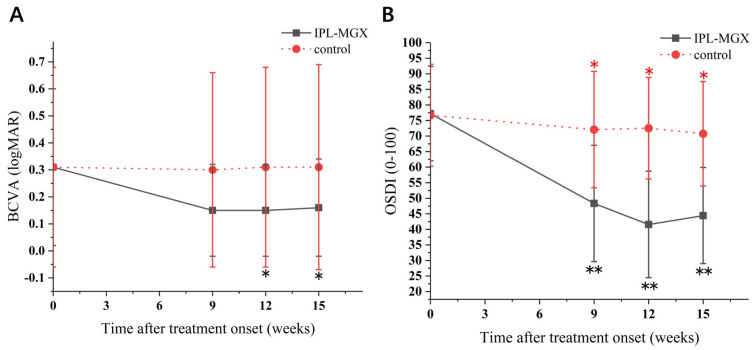
Time course of Snellen best–corrected visual acuity (BCVA) and Ocular Surface Disease Index (OSDI) score in the intense pulsed light (IPL)–meibomian gland expression (MGX) and control groups. (**A**) Changes of BCVA before and after treatment. (**B**) Changes of OSDI score before and after treatment. All follow–up assessments were compared to baseline (time 0) values (* *p* < 0.05, ** *p* < 0.001).

**Figure 3 jcm-11-01377-f003:**
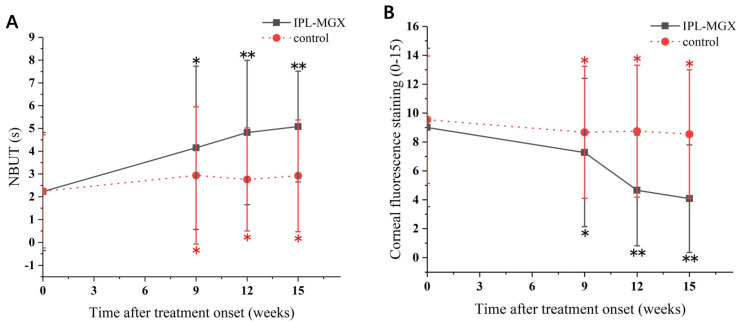
Time course of non–invasive tear breakup time (NBUT) and corneal fluorescein staining (CFS) in the intense pulsed light (IPL)–meibomian gland expression (MGX) and control groups. (**A**) Changes of NBUT before and after treatment. (**B**) Changes of CFS before and after treatment. All follow–up assessments were compared to the baseline (time 0) values (* *p* < 0.05, ** *p* < 0.001).

**Figure 4 jcm-11-01377-f004:**
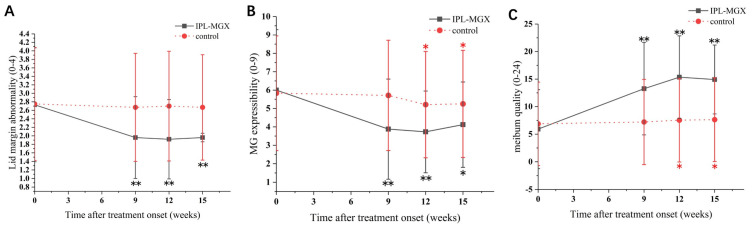
Time course of eyelid margin abnormalities, meibomian gland expressibility (MGX), and meibum quality in the intense pulsed light (IPL)–meibomian gland expression (MGX) and control groups. (**A**) Changes of lid margin abnormality before and after treatment. (**B**) Changes of MGX before and after treatment. (**C**) Changes of meibum quality before and after treatment. All follow-up assessments were compared to the baseline (time 0) values (* *p* < 0.05, ** *p* < 0.001).

**Figure 5 jcm-11-01377-f005:**
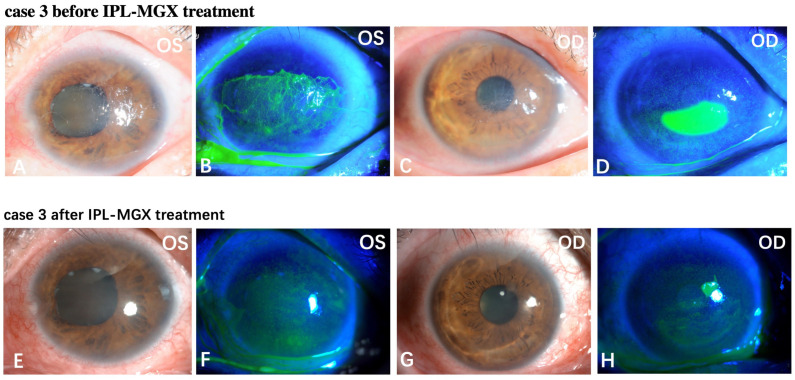
Case 3 in the IPL-MGX group exhibited considerable improvement in the ocular surface condition from baseline to the last follow-up assessment. Images of the ocular surface and corneal fluorescein staining of both eyes at baseline (**A**–**D**) and during the last follow-up visit (**E**–**H**).

**Figure 6 jcm-11-01377-f006:**
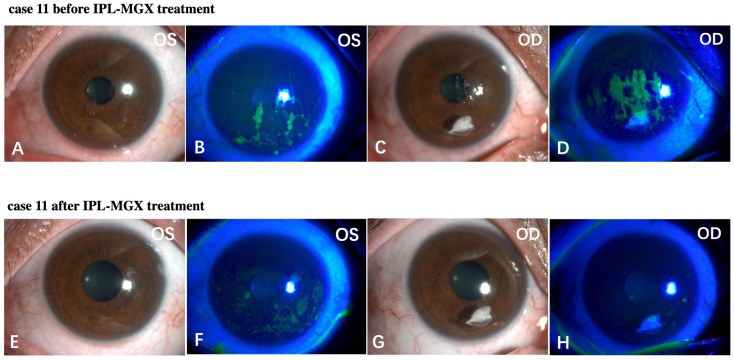
Case 11 in the IPL-MGX group exhibited considerable improvement in the ocular surface condition from baseline to last follow-up assessment. Images of the ocular surface and corneal fluorescein staining of both eyes at baseline (**A**–**D**) and during the last follow-up visit (**E**–**H**).

**Table 1 jcm-11-01377-t001:** Characteristics of the study subjects in the intense pulsed light (IPL)-meibomian gland expression (MGX) and control groups.

Characteristic		IPL-MGX Group (*n* = 26)	Control Group (*n* = 24)	*p*-Value
Age (year), mean SD (range)		53.46 ± 10.71 (32–78)	51.71 ± 13.32 (30–67)	0.861
Sex (male/female)		1/25	0/24	1.000
History of DE (years), mean SD (range)		6.38 ± 4.34 (1–14)	6.92 ± 5.14 (1–16)	0.792
History of SS (years), mean SD (range)		8.19 ± 4.72 (1–17)	8.29 ± 5.38 (1–20)	0.961
Refractory SS-DE		23 (88.46%)	21 (87.5%)	1.000
Previous treatments	Lubricant eyedrops or ointment	26	23	0.968
	warm compress and massage	23	22	1.000
	Topical anti-inflammatory	19	17	0.860
	Contact lenses	1	1	1.000
	Moisture chamber spectacles	2	1	1.000
	Punctal plugs	1	1	1.000
	Systemic immunosuppression	14	11	0.571

SS: Sjögren’s syndrome, DE: dry eye.

**Table 2 jcm-11-01377-t002:** Characteristics of the intense pulsed light (IPL)-meibomian gland expression (MGX) and control groups before and after treatment.

Characteristic	Group	BL		9 Weeks after Treatment Onset			12 Weeks after Treatment Onset			15 Weeks after Treatment Onset		
		Mean ± SD	*p*-Value for	Mean ± SD	*p*-Value vs.	*p*-Value for	Mean ± SD	*p*-Value vs.	*p*-Value for	Mean ± SD	*p*-Value vs.	*p*-Value for
			IPL-MGX vs. Control		BL	IPL-MGX vs. Control		BL	IPL-MGX vs. Control		BL	IPL-MGX vs. Control
BCVA (logMAR)	IPL-MGX group	0.31 ± 0.29	0.625	0.15 ± 0.17	0.001	0.352	0.15 ± 0.17	<0.001	0.341	0.16 ± 0.18	<0.001	0.352
	control	0.31 ± 0.37		0.30 ± 0.36	0.109		0.31 ± 0.37	0.581		0.31 ± 0.38	1.000	
OSDI (0–100)	IPL-MGX group	77.20 ± 15.18	0.934	48.34 ± 18.70	<0.001	<0.001	41.57 ± 17.14	<0.001	<0.001	44.42 ± 15.44	<0.001	<0.001
	Control	76.68 ± 16.39		72.07 ± 18.70	0.002		72.53 ± 16.31	0.028		70.74 ± 16.80	0.002	
Conjunctival congestion	IPL-MGX group	1.82 ± 0.77	0.831	1.65 ± 0.58	0.074	0.606	1.55 ± 0.56	0.005	0.163	1.53 ± 0.58	0.002	0.227
	Control	1.80 ± 0.66		1.75 ± 0.64	0.087		1.80 ± 0.67	0.885		1.75 ± 0.62	0.091	
TMH (mm)	IPL-MGX group	0.13 ± 0.04	0.453	0.14 ± 0. 04	0.253	0.869	0.16 ± 0.04	0.013	0.453	0.15 ± 0.03	0.037	0.837
	Control	0.14 ± 0.04		0.14 ± 0.04	0.445		0.15 ± 0.04	0.005		0.15 ± 0.04	<0.001	
SIT (mm/5min)	IPL-MGX group	3.42 ± 2.80	0.929	4.27 ± 2.99	0.248	0.337	3.85 ± 1.74	0.201	0.798	4.62 ± 2.45	0.024	0.306
	Control	3.62 ± 3.00		3.75 ± 3.34	0.718		4.20 ± 3.37	0.054		4.08 ± 3.27	0.069	
NBUT (s)	IPL-MGX group	2.23 ± 2.59	0.800	4.15 ± 3.58	0.001	0.551	4.82 ± 3.17	<0.001	0.009	5.08 ± 2.433	<0.001	0.006
	Control	2.24 ± 2.48		2.94 ± 3.01	0.001		2.76 ± 2.26	0.019		2.92 ± 2.45	0.007	
Corneal fluorescence staining (0–15)	IPL-MGX group	9.00 ± 5.49	0.822	7.27 ± 5.13	0.001	0.399	4.65 ± 3.83	<0.001	0.001	4.08 ± 3.72	<0.001	0.001
	Control	9.54 ± 4.41		8.67 ± 4.57	0.002		8.75 ± 4.57	0.003		8.54 ± 4.46	0.003	
MG dropouts (0–3)	IPL-MGX group	1.58 ± 1.24	0.502	1.46 ± 1.217	0.257	0.283	1.46 ± 1.17	0.257	0.283	1.46 ± 1.17	0.257	0.283
	control	1.79 ± 1.10		1.79 ± 1.10	1.000		1.79 ± 1.10	1.000		1.79 ± 1.1	1.000	
Lid margin abnormality (0–4)	IPL-MGX group	2.73 ± 1.34	0.968	1.96 ± 0.96	<0.001	0.026	1.92 ± 0.93	<0.001	0.012	1.96 ± 0.10	<0.001	0.019
	control	2.75 ± 1.33		2.67 ± 1.27	0.317		2.70 ± 1.29	0.083		2.67 ± 1.24	0.157	
MG expressibility (0–9)	IPL-MGX group	6.00 ± 3.29	0.733	3.88 ± 2.72	<0.001	0.026	3.73 ± 2.22	<0.001	0.040	4.12 ± 2.32	0.002	0.094
	control	5.83 ± 3.12		5.71 ± 3.00	0.429		5.21 ± 2.89	0.001		5.25 ± 2.91	0.003	
meibum quality (0–24)	IPL-MGX group	5.92 ± 7.16	0.360	13.26 ± 8.40	<0.001	0.003	15.35 ± 7.49	<0.001	<0.001	14.92 ± 6.25	<0.001	0.001
	control	6.88 ± 7.57		7.21 ± 7.73	0.114		7.54 ± 7.58	0.013		7.63 ± 7.60	0.007	

BCVA: best corrected visual acuity, OSDI: ocular surface disease index, TMH: tear meniscus height, NBUT: non-invasive break up time of tear film, SIT: Schirmer I test, MG: meibomian gland.

## Data Availability

Individual deidentified participant data (including data dictionaries) will be shared. All the individual participant data collected during the trial, after deidentification, Study Protocol, Statistical Study Protocol, Statistical Analysis Plan, Analytic Code, and Study Protocol will be available beginning 6 months and ending 1 year following article publication to anyone who wishes to access the data to achieve aims in the approved proposal. Proposals should be directed to lzyjxm@zju.edu.cn. Data requestors will need to sign a data access agreement.

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
