# Peer review of "Therapeutic Effect of Intense Pulsed Light in Patients with Sjögren’s Syndrome Related Dry Eye"

_jcm, 2022, doi:10.3390/jcm11051377_

Round 1
Reviewer 1 Report
The topic seems very important and useful for clinicians, however, there were several major concerns in this manuscript as below.
- First of all, therapeutic effect of IPL on SS has been already published by Arita, et al. Multicenter Study of Intense Pulsed Light for Patients with Refractory Aqueous-Deficient Dry Eye Accompanied by Mild Meibomian Gland Dysfunction.
Author Response
Reply1: Thank you very much for your reminder. We have rewritten our introduction, discussion (third paragraph) and refer that paper. That was a small sample retrospective study containing 7 SS-DE patients in IPL-MGX and 6 SS-DE patients in control groups. All patients were allowed to various eye drops during the whole therapy and both eyes of each participant were included.
Reply2: Thank you very much. Compared to non-SS DE patients, the SS-DE patients significantly had more serious signs and symptoms, especially those moderate to severe ones. it is very difficult for them to stop using daily artificial tears. To protect the patients interests and prevent too many patients withdraw from the study, especially from control group, we allowed them to use artificial tears as basic treatment in both IPL-MGX and control groups.
Reply3: Thank you very much. It is a good suggestion. We add more discuss on the mechanism in Discussion (paragraph 1 and 5)
Reply4:Thank you very much. Home-care such as warm compresses, lid hygiene was allowed to continue in the study. I think it might be the reason that MG expressibility and quality improved.
Reviewer 2 Report
Dear authors, thank you very much for sending your work to this journal, this is a very interesting study about a pathology that we usually struggle to treat, and IPL opens a new way to explore in patients with different problems affecting their ocular surface. I have a few comments:
- Methods (Experiment Design): it seems that the term IPM-MGX is not correct and should be amended.
- Why did you consider studying only patients with refractory disease?
- Page 3, 2.4.3. paragraph there is a spelling mistake in "Oclus", it should be "Oculus"
- In the results section, first paragraph, could you explain a bit more what happened with that patient that leave the study in the IPL-MGX group?
- In the last paragraph of the discussion, you state " Unexpectedly, the vision of the SS-DE patients showed a remarkable improvement
after IPL-MGX treatment at each follow-up". Why is this finding unexpected for you? it is well known that severe dry eye causes vision abnormalities, in my opinion, it is clearly expected that an improvement in the overall homeostasis of the ocular surface might be associated with an improvement in visual acuity. Please comment on that. - In the conclusion section, you mention that this treatment might be especially beneficial for refractory patients, considering that you only use patients with refractory disease, it might be ventured saying such formation. Do you think we should study first the effect of the treatment in not that severe patients? please comment.
Author Response
Reply 1: Thank you very much. The mistake has been corrected.
Reply 2: Thank you very much. We chose refractory SS-DE based on the following reasons. (1) Artificial tear would be sufficient in most mild SS-DE patient. However, advanced SS-DE patients might not respond to standard therapies, we want to find a new, powerful and safe therapy for those refractory SS-DE patients. (2) There are multiple studies proved that IPL is effective for improvement of meibomian gland dysfunction in MGD patients, however, the effect on lacrimal gland secretion is unclear. Most mild SS-DE might only present with decreased lacrimal gland secretion function, but have normal meibomian gland function and lack of typical keratoconjunctivitis sicca phenotypes. (more details in our previous study https://doi.org/10.3389/fmed.2019.00291). We found that the lacrimal glands receive a greater influence in mild SS-DE patients, and in those advanced and refractory SS-DE patients, meibomian glands are greatly affected. That’s why we decided to evaluate the efficacy and safety of the IPL-MGX treat in SS patients.
Reply 3: Thank you very much. The spelling mistake has been corrected.
Reply 4: Thank you. That patient was not able to follow-up as required due to the COVID-19 quarantine policy. The situation has been stated in the text.
Reply 5: Thank you very much. We didn’t observe vision improvement in MGD related dry eye patient in our and many others’ IPL-MGX studies. The reason might be that MGD patients usually don’t present such a bad ocular surface condition, so their vision acuity is normally not affected. Vision acuity is widely used as safety assessment in almost all the IPL-MGX experiments (to prove that IPL treatment did not damage cornea, lens or retina). The improvement in this study, was beyond our expectation.
Reply 6: Thank you very much. It is a good question. As stated earlier, artificial tear or other simple home-care such as warm compresses, lid hygiene might be sufficient in most of the mild SS-DE patients. In clinical practice, we have treated a large number of SS-DE patients, including those mild SS-DE, which have achieved good treatment results. When we design this study, we thought choosing severe cases might be more persuasive to prove that IPL can not only for MGD patients, but also for SS patients.
Reviewer 3 Report
I am very grateful to be able to read and review such a great work.
The authors described an interesting, comprehensive and informative work focused on possible usage of the IPL in the Sjoegren syndrome patients compared to the control group. The proper sample size analysis was conducted and presented accordingly to the best scientific practice. Authors cover a very important and clinically significant part of everyday life in the corneal service.
Clear and focused research questions have been stated in a proper and easy to revise manner. The text is concise and can be read with ease. Only some minor revisions are needed:
- Please provide the primary and secondary endpoints or outcome measures. Preferably clearly stated as they can be easily reviewed and found by readers.
- Please explain why for fluorescein staining no verified staining classification was used? There are Oxford Scale, NEI scale, SICCA Scale some are dedicated for Sjoegren syndrome, and the authors still did not used any of those, why? If you used a non-verified, accepted worldwide staining scale please provide the representative cases as an identification of each stain or a drawing of such.
- Similarly, in the subsection 2.4.7 please provide the representative cases of each abnormality if you use a scale that has not been described previously.
- The figure 2, 3 and 4 are of poor quality – please provide better graphs (the scale - numbers are not visible and red dots are a bit small – difficult to see even in the screen).
- Please provide the limitation of your study with proper defence. I am aware that as an author’s this could be difficult but also express that you can be critical toward your own work and will help you improve the current paper as well as provide scientific direction for future work.
Author Response
Reply 1: Thank you very much. It is a good suggestion. We have added relevant information in 2.2 Experiment design.
Reply 2: Thank you. We used NEI scale which was published by Lemp in 1995. The 5 corneal zones scale was developed by Caffery and Josephson in 1991 and now widely distributed and adapted by the Brien Holden Vision Institute (www.brienholdenvision.org). The National Eye Institute (NEI)/Industry scale continued the Caffery and Josephson concept of grading 5 corneal zones, and scored the zones by the density of stained dots on a 0-3 scale. This 5 corneal zone scale was devised for varying clinical populations, including Sjoegren syndrome. More details could be found in “Ocul Surf 2019 Apr;17(2):208-220. doi: 10.1016/j.jtos.2019.01.004. Epub 2019 Jan 14.”
Reply 3: Thank you. Both scales for corneal fluorescein staining and lid margin abnormalities we used were reported in our previous study before (Ann Transl Med. 2021 Feb;9(3):238. doi: 10.21037/atm-20-1745.). The scale for lid margin referred to the Zhang’s protocal (XiaoZhao Zhang, Nan Song & Lan Gong (2019) Therapeutic Effect of Intense Pulsed Light on Ocular Demodicosis, Current Eye Research, 44:3, 250-256, DOI: 10.1080/02713683.2018.1536217).
Reply 4: Thank you very much. The figures have been modified and replaced.
Reply 5: Thank you very much, we appreciate. We have added our limitations in 4. Discussion (last paragraph).
Round 2
Reviewer 1 Report
The authors amended well.